# A ‘Hybrid’ Radiotherapy Regimen Designed for Immunomodulation: Combining High-Dose Radiotherapy with Low-Dose Radiotherapy

**DOI:** 10.3390/cancers14143505

**Published:** 2022-07-19

**Authors:** Hongshan Ji, Zhiguo Zhou

**Affiliations:** Department of Radiation Oncology, The Fourth Hospital of Hebei Medical University, Shijiazhuang 050000, China; hongshanji0731@126.com

**Keywords:** high-dose radiotherapy, low-dose radiotherapy, immune, radiotherapy dose, fraction, treatment volume, multisite radiotherapy

## Abstract

**Simple Summary:**

Radiotherapy is an important cancer treatment. Aside from its direct killing effect, it also affects anti-tumor immunity. However, radiotherapy’s immune effect is not clear; it depends on the dose and fraction, cancer type, combined immunotherapy, and many other factors. Studies have focused on the optimal radiotherapy regimen to stimulate anti-tumor immunity, but conflicts exist, especially regarding the best radiation dose and fractions. Interestingly, high-dose radiotherapy and low-dose radiotherapy have complementary effects on stimulating anti-tumor immunity. Preclinical studies supporting this finding have accumulated, but gaps between theory and clinical practice still exist. This review summarizes the evidence supporting the use of this ‘hybrid’ radiotherapy approach to effectively stimulate anti-tumor immunity, explains the immune mechanisms of this combination, raises questions that must be addressed in clinical practice, and provides ideas for designing individualized treatment to increase efficiency in stimulating anti-tumor immunity using high-dose plus low-dose radiotherapy.

**Abstract:**

Radiotherapy (RT) affects anti-tumor immunity. However, the exact impact of RT on anti-tumor immune response differs among cancer types, RT dose and fractions, patients’ innate immunity, and many other factors. There are conflicting findings on the optimal radiation dose and fractions to stimulate effective anti-tumor immunity. High-dose radiotherapy (HDRT) acts in the same way as a double-edged sword in stimulating anti-tumor immunity, while low-dose radiotherapy (LDRT) seems to play a vital role in modulating the tumor immune microenvironment. Recent preclinical data suggest that a ‘hybrid’ radiotherapy regimen, which refers to combining HDRT with LDRT, can reap the advantages of both. Clinical data have also indicated a promising potential. However, there are still questions to be addressed in order to put this novel combination therapy into clinical practice. For example, the selection of treatment site, treatment volume, the sequencing of high-dose radiotherapy and low-dose radiotherapy, combined immunotherapy, and so on. This review summarizes the current evidence supporting the use of HDRT + LDRT, explains possible immune biology mechanisms of this ‘hybrid’ radiotherapy, raises questions to be considered when working out individualized treatment plans, and lists possible avenues to increase efficiency in stimulating anti-tumor immunity using high-dose plus low-dose radiotherapy.

## 1. Introduction

Radiotherapy (RT) is a powerful strategy for activating anti-tumor immunity through various mechanisms [1]. It plays crucial roles in regulating the tumor immune microenvironment (TME), shifting it towards an immune-favorable type, and amplifying the immunotherapy effect [2]. However, the immune-stimulating effect of RT seems to be uncertain, depending on the cancer type, RT dose and fractions, combined immunotherapy, and many other factors [3]. Of particular concern is the best RT dose and fractions to stimulate anti-tumor immunity and overcome the immune-suppressive barriers of the TME, which is still the main challenge that is faced by many researchers and clinical workers.

In recent years, many studies have investigated the immune-stimulating effect of different RT doses on different cancer types [4,5,6]. It is now well established that high-dose radiotherapy (HDRT), defined as more than 5 Gy/fraction, can trigger many pathways to activate the innate or adaptive immune system against tumors [7]. HDRT can induce an in situ vaccination effect, modify the phenotype of tumor cells to render them more ‘visible’ to T-cell killing, and alter the tumor microenvironment to promote greater infiltration of immune effector cells [6,8,9,10,11]. However, HDRT also has negative effects on anti-tumor immunity, such as recruiting immune-suppressive cells and increasing immune-regulatory cytokines secretion [12]. Furthermore, radiation, especially when the dose is high, has a cytotoxic effect on anti-tumor immune cells, which will impair anti-tumor immunity. Researchers have spent much effort on optimizing HDRT’s immune-stimulating effect and attenuating its negative immune effects [13,14,15]. Meanwhile, the evidence of the immune-modulating effect of low-dose RT (LDRT), defined as lower than 2 Gy/fraction, is also emerging [16,17]. Interestingly, HDRT’s and LDRT’s effects on anti-tumor immunity seem to be complementary in many pathways. In contrast to HDRT, LDRT plays a critical role in modulating the tumor immune microenvironment via enhancing immune effector cell infiltration and attenuating the immune-suppressive effects of RT [18]. Thus, using a ‘hybrid’ RT regimen may be the way to optimally stimulate anti-tumor immunity.

This combination of RT doses has been tested in some preclinical and clinical studies, yielding hopeful results [19,20,21]. However, research to date has not yet solved some aspects that must be considered when putting this novel RT treatment regimen into clinical practice. The main question is how to combine high- and low-dose RT in space and time. For example, when treating a metastatic tumor, we must decide which tumor site should be treated with HDRT or LDRT. Whether we should deliver HDRT and LDRT to the same tumor site and what the proper sequence of HDRT and LDRT should be are questions that need to be discussed and explored in future studies. This review summarizes the current knowledge and promising investigations into inducing anti-tumor immunity using HDRT + LDRT and provides ideas for combining HDRT + LDRT in space and in time.

## 2. HDRT + LDRT in Stimulating Anti-Tumor Immunity: A Preclinical Snapshot

In 2020, Barsoumian et al. established an RT regimen that was called the ‘RadScopal Technique’, which referred to treating the primary tumor with HDRT (12 Gy*3) and secondary metastases with LDRT(1 Gy*2) 3 days later. They combined this new approach with immune checkpoint inhibitors (ICIs) in mice bearing 344SQ lung adenocarcinoma. They observed that the HDRT + LDRT + ICIs group showed slower primary and secondary tumor growth, higher levels of immune effector cell infiltration, and a greater reduction of transforming growth factor beta (TGF-β) at secondary sites than mice that were receiving HDRT or LDRT alone or in combination with immunotherapy [19]. TGF-β is a powerful immunosuppressive cytokine that hinders the cross-priming of T-cells, impairs T-cells functional differentiation, and recruits Tregs [22,23]. This phenomenon suggested that HDRT + LDRT was superior to HDRT or LDRT alone in tumor control and activation of anti-tumor immunity.

More recently, the same group combined the ‘RadScopal Technique’ with other immunotherapy and observed similar results [20,24,25]. Aside from applying HDRT to the primary tumor and LDRT to the secondary tumor, other researchers have applied LDRT to a larger treatment volume such as a whole metastatic organ or the entire body, harvesting similar immune-modulating effects [16,26]. A novel systematic low-dose RT technique, targeted radionuclide therapy (TRT), has also been proven to exert satisfying immune-stimulating effects in combination with HDRT [27]. The details of these studies are provided in Table 1.

## 3. Immunobiology Mechanisms of HDRT + LDRT

HDRT and LDRT have distinct effects on inducing anti-tumor immunity. HDRT is likely to ‘inspire’ the immune system via various immune pathways that trigger recognition and presentation of tumor-associated antigens. However, HDRT itself is not without drawbacks. It has the unwanted side effect of dampening later immune responses in a so-called ‘rebound’ immune suppression [28]. HDRT recruits immuno-suppressive cell populations such as Tregs, myeloid-derived suppressor cells (MDSCs), and tumor-associated macrophages (TAMs) [29]. In contrast, LDRT polarizes TAMs to the anti-tumor M1 phenotype, reduces immune-suppressive MDSCs and Tregs, and improves T-cell infiltration [30,31]. To illustrate this complementary effect of HDRT and LDRT, we summarize evidence that explains the immunobiology mechanism of HDRT and LDRT and establishes a hypothesis-based immune response model that shows the immune pathways that are activated after HDRT + LDRT (Figure 1).

### 3.1. HDRT: A Powerful Tool to Inspire Anti-Tumor Immunity

HDRT plays powerful roles in achieving >90% local control [32] and inducing immunogenic cell death (ICD) [33] with an increase in antigen presentation [9]. HDRT kills tumor cells directly, and the tumor cell debris can be taken up by antigen-presenting cells (APCs) such as dentric cells (DCs). Another important mechanism is triggering the cyclic GMP-AMPsynthase (cGAS)- stimulator of interferon genes (STING)-IFN-1 pathway [34,35,36]. Type I interferon (IFN-1) recruits DCs that cross-prime CD8+ T-cells against tumor antigens [37]. DCs that are activated by the two mechanisms that are described above then migrate to the lymph nodes and participate in the cross-priming of naïve CD8 + T-cells [38]. The activated tumor-specific CD8 + cytotoxic T-cells enter the blood flow and infiltrate both the irradiated tumor and the non-irradiated lesion, eliminating tumor cells. This process is the well-known ‘abscopal effect’ that was first described in 1953 by Mole et al., referring to systemic immune responses that are triggered following radiotherapy that is applied to a local tumor site [39].

In addition, HDRT is associated with the strong release of damage-associated molecules (DAMPs), further promoting an immune response following RT [40,41]. DAMPs, mainly including calreticulin (CRT), heat-shock proteins (HSP70 and HSP90), adenosine triphosphate (ATP), and high-mobility group box-1 (HMGB1), are molecules that are released by dying or stressed cells that function as either adjuvants or danger signals for the immune system [42]. These molecules trigger different pathways to activate the anti-tumor immune system. For example, HMGB1 promotes tumor antigen presentation [43], recruits inflammatory cells, and mediates interactions between natural killer cells (NKs), DCs, and macrophages [44]. CRT acts as a pro-phagocytic signal by binding to the CD91 receptor on macrophages and DCs, promoting tumor antigen presentation [45]. ATP-P2X7 purinergic receptor (P2X7R) signaling enhances immune cell recruitment [46].

HDRT induces tumor cells to upregulate several cell surface molecules, including major histocompatibility complex (MHC) class 1 [9], the apoptosis-inducing death receptor FAS, and several natural killer group 2D (NKG2D) ligands, which enhance the recognition and cytolysis of the tumor by T-cells and NKs, respectively [47]. HDRT increases pro-inflammatory cytokines and chemokines such as CXCL10 and CXCL16, which promotes effector immune cell infiltration [48,49]. High-dose irradiation upregulates IL-1β, TNF-α, and Type 1 and 2 interferons [50], which upregulate intracellular adhesion molecule-1 (ICAM-1) and vascular adhesion molecule-1 (VCAM-1) on the tumor endothelium [51,52]. These cell adhesion molecules promote migration of lymphocytes into the tumor parenchyma.

More importantly, all the pro-immune effects that are described above are associated with a relatively higher dose of RT rather than LDRT, indicating that HDRT has its unique role in activating anti-tumor immunity [9,41,53,54,55,56,57]. A higher dose of radiation is more likely to induce immunogenic cell death that is mediated by DAMPs, activate cGAS-STING-dependent IFN-1 production, and upregulate pro-inflammatory molecules, providing the first underpinning of combining HDRT with LDRT.

### 3.2. LDRT: An Immune Modulator

Several immunosuppressive features can preclude HDRT induction of anti-tumor immunity [22,58,59]. Although LDRT is not as effective as HDRT in stimulating the anti-tumor immune response, its immune-modulating effect can attenuate the side effects of HDRT that impair anti-tumor immunity, acting as a complementary product in the combination treatment.

Firstly, HDRT recruits many immune-suppressive cells such as MDSCs, Tregs, M2 macrophages, and cancer-associated fibroblasts (CAFs); causes the release of immune-suppressive factors; and dampens anti-tumor immunity [12,14,60,61,62]. M2 macrophages secrete immunosuppressive mediators such as IL-10 and TGF-β, inhibiting anti-tumor immunity and promoting a radioresistant phenotype [63,64]. Tregs cause CTLA-4 expression, IL-10 and adenosine release, leading to the inhibition of T-cell activation and poor outcome [65]. Additionally, MDSCs inhibit T-cell function and anti-tumor immunity through arginase-mediated arginine depletion, nitric oxide production, and reactive oxygen species release [66]. CAFs play a powerful role in immunosuppression via the secretion of relevant immunosuppressive molecules such as prostaglandin E2, interleukin-6 and 10, or TGF-β, resulting in impaired T-cells activation and DC maturation, and this effect remains unchanged after HDRT [59].

In contrast, LDRT reduces negative regulatory factors and shifts immune cell subpopulations to favor tumor control [17,19,67]. Liu et al. observed that total body low-dose irradiation (TBI) of 1.25 Gy significantly decreased the number of Tregs, while increasing the effector-memory T-cells [67]. Klug et al. found that LDRT (2 Gy*1) reprogrammed the TME by polarizing iNOS+ M1 macrophages. In turn, iNOS activity by these reprogrammed macrophages was responsible for vascular normalization and activation, T-cell recruitment, and tumor rejection [31]. A Phase 2 randomized study showed patients that were receiving HDRT (24 Gy in 3 fractions) experienced a decrease in the ratio of M1/M2 macrophages in the TME, while subjects that were receiving LDRT (2 Gy in 4 fractions) experienced the opposite effects [68].

LDRT also attenuates other immune-suppressive effects of HDRT. HDRT negatively affects the tumor immune microenvironment by damaging the tumor vasculature [69,70,71] and reducing blood flow, which preclude immune cell infiltration and exacerbates the hypoxia-driven immunosuppressive environment [58,72,73]. In contrast, LDRT promotes immune effector cell infiltration, acting as a ‘TME modulator’ [18]. Dovedi et al. demonstrated that low-dose fractionated radiotherapy (5*2 Gy) enhanced T-cell trafficking to irradiated tumor sites and augmented resident anticancer T-cell responses with the capacity to mediate the abscopal effect [74]. Herrera et al. also demonstrated that a dose of 0.5 to 1 Gy enhanced T-cell infiltration and rendered immune desert tumors responsive to its combination with immunotherapy. Furthermore, they translated the preclinical findings to the clinic with a pilot study in eight patients with metastatic immune cold tumors, yielding a response rate of 12.5% [17].

In total, LDRT complements the immune effects of HDRT. LDRT increases immune system access to the TME and mitigates the immunosuppressive consequences of HDRT.

## 4. Clinical Challenges: How Will We Bridge the Gap between Theory and Practice?

Although the theory of combining HDRT + LDRT has matured, there are still gaps between the theory and clinical practice. Some researchers have brought this combination RT into the clinic with the aim of improving outcomes [75]. However, positive results are still rare. There are still many questions that need further investigation.

Patel et al. conducted a prospective Phase II trial of HDRT with or without LDRT for metastatic cancer [21]. Disease control rate (DCR) was defined as a complete/partial response [CR/PR] or stable disease [SD], and overall response rate (ORR) was defined as CR/PR at any point. The four-month DCR was 47% [16/34] in the HDRT + LDRT group vs. 37% [14/38] in the HDRT alone group, *p* = 0.38, and ORR was 6% [9/34] in the HDRT + LDRT group vs. 13% [5/38] in the HDRT alone group, *p* = 0.27. Although tumor response did not meet statistical significance, the LDRT lesion response (53%) improved compared to the nonirradiated lesions in the HDRT + LDRT (23%, *p* = 0.002) and HDRT (11%, *p* < 0.001) groups. LDRT enhanced T-cells and NK cell infiltration in irradiated lesions.

Several factors contributed to the lack of statistical significance in this trial. (i) The choice of radiation, dose, fractionation, and the number of sites lacked randomization owing to consideration of treatment efficiency and safety. (ii) The number of patients that were enrolled was limited. Here, we raise questions needing further investigation and provide suggestions on how to utilize HDRT + LDRT in the clinic with respect to patients’ immune functions, disease burden, and other factors.

### 4.1. How to Decide the Number of HDRT Site

Theoretically, multi-site HDRT is superior to the single-site strategy in activating systemic immune response [76]. Each tumor lesion provides an independent release of distinct tumor-associated antigens (TAAs), which cannot be shared by all the metastatic sites [77]. Multiple site HDRT might circumvent the issue of tumor heterogeneity by priming ‘visualization’ of a wider range of TAAs. Furthermore, all-site irradiation can stimulate the tumor vasculature and negate the immunosuppressive features of bulky disease, thus enhancing immune cell penetration in all tumor lesions [78,79,80]. In addition, delivering SBRT to all tumor sites directly ensured tumor sterilization.

A Phase I trial that was conducted by Luke et al. demonstrated the feasibility of multisite stereotactic body radiotherapy (SBRT) and pembrolizumab in patients with advanced solid tumors [81]. They observed that the overall objective response rate was 13.2% and the out-of-field response rate (CR/PR) of nonirradiated target metastasis was 26.9%. The median overall survival was 9.6 months and the median progression-free survival was 3.1 months. These data showed that multisite SBRT effectively controlled both the primary and distant tumor with acceptable toxicity in patients with metastatic cancer. Moreover, a secondary analysis of this trial showed increased expression of innate and adaptive immune genes after SBRT [82]. In conclusion, as long as it is feasible and tolerable, HDRT should be given to as many tumor sites as possible to achieve efficient tumor control and systemic immune response. For one thing, delivering HDRT to all tumor sites ensures tumor sterilization; for another, multisite HDRT is prone to activate systemic immune response, which is crucial for eliminating circulating tumor cells that cannot be eliminated by HDRT.

However, multisite HDRT has drawbacks. One of the biggest challenges is that we lack knowledge of the appropriate organ dose–volume constraints when high RT doses are delivered to multiple isocenters [83]. In plans with multiple isocenters, the scatter dose and low-dose bath are likely to affect the circulating immune cells, particularly highly radiosensitive lymphocytes, leading to T-cell depletion, impairing anti-tumor immunity, and reducing response to ICIs. Indeed, replacing HDRT at some sites with LDRT may offer a solution to this problem, but toxicities that are related to low-dose irradiation of large tumor volumes is yet to be defined.

Aside from intentionally delivering LDRT to large tumor sites, scattered low doses can also act as an immune modulator if a large-volume tumor lesion is close to the lesion that is treated with HDRT. Welsh et al. observed in a Phase II trial that when treating patients bearing metastatic tumors with SBRT, non-targeted lesions that unintentionally received low-dose radiation were more likely to respond than those that received no radiation (31 vs. 5%, *p* = 0.0091) [84]. A post hoc analysis that was conducted by Menon et al. had similar results. Their data indicated that 58% of the lesions that unintentionally received scattered low-dose irradiation met the PR/CR criteria for RECIST, compared with 18% of the lesions that received no dose (*p* = 0.001) [85]. Consequently, scattered low-dose irradiation can be potentially utilized to enhance the abscopal effect and tumor control.

Another challenge of multisite HDRT is that adding to the number of tumor sites that are receiving HDRT may aggravate its immune-suppressive effect. Schoenhals et al. found that delivering HDRT to both tumors accelerated the tumor growth, which could be attributed to HDRT’s ability to induce higher levels of Tregs. Adding anti-Treg immunotherapy abrogated this immune-suppression effect and the abscopal effect was restored [86]. In this experiment, the researchers used anti-Treg immunotherapy to overcome the immune-suppressive effects of HDRT. However, we argue that LDRT or immunotherapy may also overcome the immune-suppressive effect of multisite HDRT. In all, immune modulation management (LDRT or anti-Treg therapy or other immunotherapy) must be implemented if we want to yield a satisfying immune-stimulating effect.

### 4.2. Target Volume of HDRT

When a tumor lesion is too large to be treated with high-dose irradiation because of toxicity, partial-volume treatment might be a choice, which has been demonstrated to have similar or even better tumor control and abscopal effect [81,87]. Luke et al.’s clinical trial made an interesting finding: patients who had at least one metastasis that was measuring >65 mL that was partially treated with SBRT had control levels that were similar to those who were treated with complete SBRT. Additionally, no significant difference in ORR, PFS, and OS was observed, indicating that the clinical responses at the irradiated site could be induced without irradiation of an entire metastasis [82].

Markovsky et al. conducted an experiment investigating whether partial-volume HDRT (10Gy*1) is inferior to full-volume radiation in tumor control in mice bearing bilateral breast tumors. Their data showed that partial HDRT led to an abscopal effect that was similar to fully exposed tumors [88]. They observed the same results when using the less immunogenic Lewis Lung Carcinoma mouse model to perform similar experiments, confirming that the findings are not unique to 67NR breast cancer or otherwise highly immunogenic tumors. Similarly, Yasmin-Karim et al. treated mice bearing bilateral prostate tumors with one site whole-tumor HDRT (5 Gy*1) or partial-volume (the center zone of tumor) HDRT. They observed when using smaller field sizes, the abscopal effect, cytotoxic CD8 + T-lymphocytes infiltration, and the positive shift of pro-inflammatory cytokines were equal to irradiating the whole tumor volume. Mice treated with partial-volume HDRT even experienced better survival [89]. These data suggest that targeting tumor sub-volumes with HDRT offers an opportunity for boosting the abscopal effect while minimizing healthy tissue toxicity. The unirradiated peripheral tumor issue will unintentionally receive LDRT, which may explain the positive immune effect, further confirming the benefit of HDRT + LDRT. More studies are needed on further optimizing the HDRT treatment parameters of this approach to boost the anti-tumor immunity and abscopal response rates with increased sparing of healthy tissue.

Another novel way of setting the target volume of HDRT has been shown to achieve strong abscopal effect. An in vitro study that was conducted by Johnsrud et al. brought spatially fractionated radiation therapy (SFRT) into view, suggesting that this form of RT delivery is superior to whole-tumor RT in inducing the abscopal effect [90]. Spatially fractionated radiation therapy (GRID) describes the delivery of a single high-dose fraction to a large treatment area that has been divided into several smaller fields with steep dose gradients, thus reducing the overall toxicity of the treatment [91]. Johnsrud et al. treated 4T1 murine breast carcinoma mice with a single dose of 20 Gy. The irradiation field covered the whole primary tumor (WTRT) or used a honeycomb beam pattern of 2 mm openings and a 4 mm center-to-center distance (GRID) to deliver 20 Gy. They observed an obvious growth inhibition of the primary tumor in mice that were receiving WTRT or GRID. However, an abscopal response was only observed in mice who received the 20 Gy GRID treatment to the primary tumor. GRID-treated mice showed greater increases in CD8+ T-cell activation and CD4+ T-cell activation levels compared to the WTRT-treated mice [90]. This form of setting the HDRT target volume is worthy of further exploration by preclinical or even clinical studies.

### 4.3. The Impact of Patients’ Immune Function

The patient’s endogenous immune system must be functioning in order to stimulate anti-tumor immunity using HDRT + LDRT. Immunotherapy-related outcomes in patients with poor immune function may be worse [92], and immune cell function is important in tumor response to radiation [12]. Some researchers divide HDRT into two kinds of doses according to their effects on the immune system: an immune-priming dose and an ablative dose [78]. The former represents a radiation dose that induces in situ vaccination. Generally, it is defined as 6–12 Gy since a dose >12–18 Gy can induce TREX1, which clears the cytosolic DNA that is induced by HDRT and consequently impairs the cGAS-STING pathway [93]. The latter is defined as a biologic effective dose (BED) in excess of 100 Gy, which is designed to ablate all tumor and immune components. Patients harboring relatively stronger endogenous immune function may be more compatible with immune-priming doses, as opposed to ablative RT. Conversely, patients with poor immune function may be less likely to benefit from multisite HDRT + LDRT. They are more suitable to receiving ablative RT to achieve ideal tumor control.

However, the particular definition of immune function remains unvalidated to date. The absolute lymphocyte count (ALC) is readily obtained from peripheral blood samples and is well standardized, which has been shown to predict clinical response to immunotherapy in several types of cancers [92,94]. Another way to stratify immune function is to characterize T-cell function and strength. For example, cell surface markers such as ICOS, GITR, OX40, 4-1BB, CD40L, and CD44 can indicate T-cell activation, while PD-1, Tim-3, Lag3, and TIGIT suggest exhaustion, or attenuated T-cell effector function [95,96,97,98,99]. However, we still lack affordable and feasible factors for quantifying the strength of an individual’s immune system.

## 5. Clinical Decision Suggestions

According to the factors that are discussed above, we offer some suggestions and possible treatment options that are based on the number of tumor lesions, patient immune status, and other factors. This hypothesis-generating decision tree is summarized in Figure 2.

### 5.1. Single Tumor Site

When treating patients with only one tumor lesion, we should pursue the ablation effect since the disease is localized. Applying LDRT after ablative HDRT may be a feasible strategy to induce anti-tumor immunity and prevent recurrence, as demonstrated in an animal experiment [26]. LDRT can be used to attenuate the immune-suppressive effect that is caused by HDRT and increase immune effector cell infiltration, providing platforms for immune checkpoint inhibitors.

### 5.2. Oligo-Metastasis (OGM) Tumor

In 1995, Hellman and Weichselbaum first proposed oligometastatic disease as a distinct cancer state between locally confined and systemic metastatic disease [100], which refers to a metastatic tumor with no more than three to five metastases [101,102,103,104,105,106]. For these patients, some researchers claim oligometastasis-directed SBRT should be delivered to all tumor lesions to yield efficient tumor control and immune-stimulating effect [76,107,108].

Over the years, some clinical evidence on the use of all-site SBRT for the treatment of oligometastatic tumors has accumulated, with efficacy and safety demonstrated [79,109]. For example, the SABR-COMET Phase II randomized trial reported that oligometastatic patients that were receiving SBRT to all tumor sites experienced a survival benefit that was comparative to patients that were receiving the best supportive care [110,111]. Another multicenter randomized Phase II trial that was conducted by Gomez et al. had similar results [112]. They found that in patients with Stage IV NSCLC and ≤3 metastatic sites, the median PFS for local consolidative therapy (SBRT or surgery) was 11.9 months vs. 3.9 months for maintenance treatment alone (*p* = 0.0054).

In conclusion, all-site SBRT might be efficient in tumor control, prolonging survival, and activating anti-tumor immunity, so we hypothesize that ablative treatments for oligometastases should be as curative as possible, and HDRT serves as a feasible choice. Patients with a relatively stronger immune system should have their endogenous immune function optimally stimulated using all-site HDRT. Immunotherapy and LDRT should be added to attenuate the immune-suppressive features of HDRT. For patients with poor immune functions, applying an immune-stimulating dose of SBRT to all tumor lesions may not elicit the ideal immune response. Given that the tumor burden is relatively small, an ablative dose should be given to all tumor sites to achieve a curative effect. However, the long-term outcomes and toxicity of all-site HDRT still need preclinical and clinical studies to explore. There might be some patients who cannot benefit from this approach. There is ongoing clinical trial testing whether HDRT + LDRT + ICI is effective in treating metastatic tumor (NCT03085719).

### 5.3. Polymetastatic Disease/High Tumor Burden

In a high tumor burden condition, it is less realistic to give HDRT to all lesions, owing to the risk of toxicity. For those with intact immune function, applying HDRT to one or a few smaller lesions to prime anti-tumor immunity, followed by LDRT to other larger lesions for stromal modulation may be beneficial. Immunotherapy should also be implemented [113]. With extensive metastasis and a high tumor burden, it is irrational to pursue an ablative effect for all tumor lesions. Therefore, the abscopal effect appears to be critical to reduce the tumor burden. Partial-volume HDRT or SFRT might serve as a choice to induce an abscopal response.

For patients with attenuated immune function and a high tumor burden, HDRT might not be suitable owing to the toxicity and the patients’ weak immune strength. Systemic therapy is still a standard treatment. Adoptive cell therapy (ACT), which is currently being studied in polymetastatic solid tumors, should be taken into consideration. A major concern for the therapeutic effect of ACT is the entry of immune cells into the immunosuppressive tumor environment. LDRT can be combined with ACT to circumvent the immunosuppressive features of the TME and enhance immune cell infiltration [114].

## 6. Conclusions

This review introduced a novel ‘hybrid’ RT regimen attempting to effectively stimulate anti-tumor immunity and exert its greatest synergy effect with immunotherapy. HDRT acts as a ‘primer’ for anti-tumor immunity, while LDRT serves as an immune modulator to overcome the immune-suppressive TME. Integrating HDRT with LDRT may be synergistic and generate early robust changes in the immune activation profile against primary and metastatic tumors. We established an immune response model showing the immune biology effect of HDRT + LDRT and built a hypothesis-based decision tree to show the possible personalized treatment choices, depending on different tumor burdens and immune function status. In addition, we raised questions that remain to be explored. Our understanding of many questions related to personalized treatment remain limited, in that little information is available. We still need more preclinical and clinical studies to explore effective way of combining HDRT with LDRT to stimulate anti-tumor immunity.

## Figures and Tables

**Figure 1 cancers-14-03505-f001:**
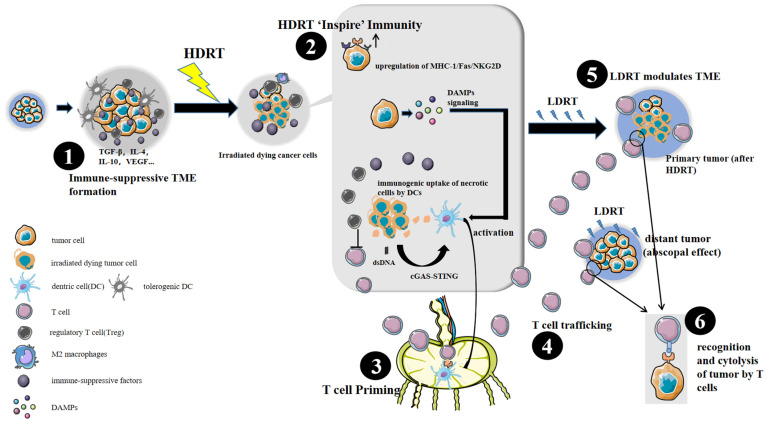
A hypothesis-based immune response model. In the course of tumor growth, the tumor microenvironment (TME) accumulates high concentrations of immune-suppressive cytokines/growth factors, such as transforming growth factor beta (TGFβ) and vascular endothelial growth factor. Under their action, immunosuppressive cells (black cells), regulatory T-cells, and myeloid cells, will prevail in the tumor environment. Dentric cells (DCs) become tolerogenic and immune effector cells cannot infiltrate the tumor (**1**). After high-dose irradiation (**2**), many immune pathways are activated. Firstly, high-dose radiotherapy (HDRT) will cause tumor cell necrosis and the release of tumor cell debris, which can be taken up by antigen-presenting cells (APCs). Secondly, HDRT causes dsDNA release, triggering the cGAS-STING-IFN-1 pathway. In addition, HDRT upregulates cell surface molecules such as MHC-1/Fas/NKG2D. Activated APCs then migrate to the lymph nodes, where they educate and prime cytotoxic T-cells (**3**). Cytotoxic T-cells will enter the bloodstream (**4**) to reach primary and distant tumor sites and kill tumor cells. However, these effector immune cells might not be able to enter primary or distant tumor stroma because of immune-suppressive barriers that are formed naturally or after HDRT. Low-dose radiotherapy (LDRT) enhances immune cells infiltration and diminishes immune-suppressive cells and cytokines, acting in a similar way to an immune-modulator (**5**). LDRT can be applied to the local tumor which has received HDRT or other tumor lesions, which remains to be explored by more studies. After LDRT, the activated immune cells successfully enter the primary and distant tumor issue, eradicating tumor cells, and causing abscopal effect (**6**). The crucial timing of LDRT needs to be explored by future studies in order to avoid effector immune cell depletion.

**Figure 2 cancers-14-03505-f002:**
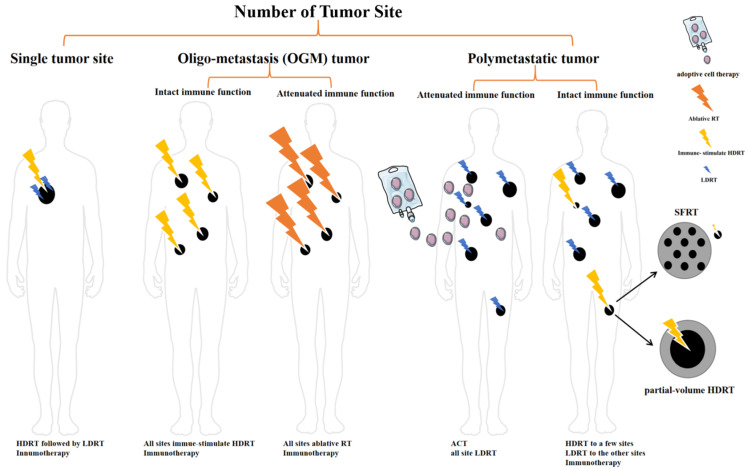
A decision tree of personalized treatment choices of patients with different tumor burden and immune status.

**Table 1 cancers-14-03505-t001:** Key preclinical studies on evaluating the valuating immune effect of high-dose radiotherapy (HDRT) + low-dose radiotherapy (LDRT).

Authors	Mice and Cell Line	Number of Tumor Sites	RT Regimen	Immunotherapy	Results
H Barsoumian et al. [19]	129Sv/Ev mice 344SQ parental lung adenocarcinoma cell line	2	12 Gy*3 HDRT to the primary tumor + 1 Gy*2 LDRT to the secondary tumor (3 days after HDRT)	anti-CTLA-4 anti-PD1	Delayed growth in both primary and secondary tumors. Enhanced natural killer cell activation, increased M1 macrophages and CD4 + T-cells, and decreased TGF-β in secondary tumors.
H Barsoumian et al. [20]	129Sv/Ev mice 344SQ parental lung adenocarcinoma cell line	2	12 Gy*3 HDRT to the primary tumor + 1 Gy*2 LDRT to the secondary tumor (3 days after HDRT)	anti-TIGIT anti-PD1	Delayed growth in both primary and secondary tumors, reduced the exhaustion of T-cells, generated effector immune memory, and prolonged survival.
Y Hu et al. [25]	129Sv/Ev mice 344SQ parental lung adenocarcinoma cell line	2	12 Gy*3 HDRT to the primary tumor + 1 Gy*2 LDRT to the secondary tumor (3 days after HDRT)	anti-PD1 anti-CTLA4 NBTXR3 nanoparticle	Slowed the growth of both primary and secondary tumors, suppressed the appearance of lung metastases, increased survival rates, induced robust long-term immune memory, and increased the CD8/Treg ratio in the secondary tumors
T Savage et al. [26]	C57BL/6 mice Lewis Lung Carcinoma, 3LL	1	22 Gy*1 + 0.5 Gy*4(12 days after HDRT) to the tumor site	-	Delayed tumor growth, increased survival, reduced Tregs and M2 macrophages in the tumor microenvironment (TME), and increased systemic T-cell responses.
	BalB/C mice breast carcinoma cell line, 4T1	1	22 Gy*1 to the tumor site + 0.5 Gy*4(12 days after HDRT) to the whole lung (metastatic prone organ)	-	Delayed local tumor progression, suppressed pulmonary metastases, remodeled the metastatic niche with decreased Tregs and increased effector T-cell infiltration in lungs, and increased survival.
J Liu et al. [16]	BALB/C mice mammary carcinoma 4T1 and colon carcinoma CT26 cell lines	2	0.1 Gy total body irradiation (3 days before HDRT) + 8 Gy*3 to the primary tumor	-	Delayed growth in both primary and secondary tumors, increased secondary tumor infiltration of CD8+ T-Cells, decreased myeloid-derived suppressor cells (MDSCs) and M2 macrophages in the secondary tumor, and inhibited metastasis
R Patel et al. [27]	C57Bl/6 and BALB/c mice B78 melanoma tumors cell line	2	targeted radionuclide therapy (TRT) using 50uCi90Y-NM600 (2.5 Gy) + 12 Gy external beam radiotherapy targeting the primary tumor	anti-CTLA4	Improved tumor response at the secondary tumor not targeted by EBRT and improved overall survival, augmented response to ICIs, and induced robust long-term immune memory.

NBTXR3: a hafnium oxide radio-enhancing nanoparticle. TIGIT: an immune checkpoint expressed on T-cells, impairing antigen presentation and T-cell proliferation. -: No immunotherapy was implemented.

## Data Availability

Not applicable.

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
