# Peer review of "A ‘Hybrid’ Radiotherapy Regimen Designed for Immunomodulation: Combining High-Dose Radiotherapy with Low-Dose Radiotherapy"

_cancers, 2022, doi:10.3390/cancers14143505_

Round 1

Reviewer 1 Report

My comments are numerous and given in the attached file. 

Author Response

Dear Editor:

Thanks for your letter and for reviewers’ comments concerning our manuscript entitled ‘A ‘Hybrid’ Radiotherapy Regimen Designed for Immunomodulation: Combining High-dose Radiotherapy with Low-dose Radiotherapy’(Manuscript ID:1774369). Those comments are all valuable and helpful for revising and improving our paper. We have studied all comments carefully and have made conscientious correction. Revised portion are marked in red in the paper. The main correction in the paper and the responds to the reviewers’ comments are as following.

Response to Reviewer 1:

We are so grateful for your comments on the manuscript. According to your advice, we amended the relevant part in the manuscript and marked them all in red. The revised manuscript is provided in the attached file.

All language errors are corrected as your request and we have edited our English using MDPI editing. Editing ID: 46425.

Comment 1: I suggest not using "hybrid", as it implies some type of multimodal approach.

Response: Thank you for your suggestions. The word ‘hybrid’ comes from another review[1] named ‘The Impact of Radiation on the Tumor Microenvironment: Effect of Dose and Fractionation Schedules’ by M Arnold et al. In this review, the authors wrote ‘Dose modulation with heterogeneous high- and low-dose regions within the tumor may provide a hybrid form of  radiation delivery that may promote the antitumor effects of both high-dose and conventional fractionation regimens.’ In this context, the word ‘hybrid’ refers to a combination form of high and low-dose radiotherapy. Other literature also used this word in the title [2,3].

[1] The Impact of Radiation on the Tumor Microenvironment: Effect of Dose and Fractionation Schedules. Cancer Growth Metastasis. 2018; 11: 1179064418761639.Published online 2018 Mar 9. doi: 10.1177/1179064418761639 PMCID: PMC5846913 PMID: 29551910

[2] Hybrid Tomo-Helical and Tomo-Direct radiotherapy for localized prostate cancer. J Appl Clin Med Phys. 2021 Oct;22(10):136-143. doi: 10.1002/acm2.13406. Epub 2021 Sep 8.

[3] Hypofractionated Hybrid Radiotherapy Techniques for Synchronous Bilateral Breast Cancer. Asian Pac J Cancer Prev. 2021 Dec 1;22(12):3933-3939. doi: 10.31557/APJCP.2021.22.12.3933.

Comment 2: Radiation will also kill anti-tumor immune cells.

Response: We add ‘Furthermore, radiation, especially when the dose is high, has a cytotoxic effect on anti-tumor immune cells, which will impaire anti-tumor immunity.’ in page 2 paragraph 2 and marked it in red.

Comment 3: line71,Not clear what is implicated by this sentence.Are the authors referring to multiple sites; primary tumor, affected nodes, metastases? Then this must be explicitly stated.

Response: In this sentence, we intended to raise some questions that clinicians might face when putting HDRT+LDRT into practice, and these questions will all be discussed in below parts. We have changed the sentence to ‘The main question is how to combine high- and low-dose RT in space and time. For example, when treating a metastatic tumor, we must decide which tumor site should be treated with HDRT or LDRT. Whether we should deliver HDRT and LDRT to the same tumor site and what the proper sequence of HDRT and LDRT should be are questions that need to be discussed and explored in future studies.’ in the third paragraph of page 2 and marked it in red.

Comment 4: Split the long sentence in two. Also, it is not clear what the observed effects are compared to conventional radiotherapy?

Response: We have corrected the sentence in the last paragraph in page 2: ‘They combined this new approach with immune checkpoint inhibitors (ICIs) in mice bearing 344SQ lung adenocarcinoma. They observed that the HDRT+LDRT+ICIs group showed slower primary and secondary tumor growth, higher levels of immune effector cell infiltration and greater reduction of TGF-β at secondary sites than mice receiving HDRT or LDRT alone or in combination with immunotherapy.’

Comment 5: Line 121: Not surprising, direct cell kill of microscopic spread.

Response: We appreciate the reviewer’s good suggestion, and we have added this point according to your ideas. We add ‘In addition, 0.5Gy*4 to the whole lung might also kill microscopic metastasis directly.’ in the third paragraph of page 3.

Comment 6: Line 124:0.1 Gy upfront is, radiobiologically, very different from 0.5x4 after, so this may well not be consistent findings. 0.5x4 may have some effect on disseminated cancer cells, while 0.1 Gy whole body should be more towards immune stimulation.

Response: It is really a good point as the reviewer suggested, and we have changed them to meet the reviewer’s thoughts in the third paragraph of page 3: ‘Unlike 0.5 Gy*4, 0.1 Gy might not be powerful enough to directly kill tumor cells. It is more likely to stimulate an anti-tumor effect.’

Comment 7:Line 155:very unclear; in what way is the stroma targeted?

Response: We have deleted ‘by targeting otherwise hostile cancer-related stroma’.

Comment 8: But a prerequisite is then that LDRT then does NOT induce TAAs? The authors must find evidence for this in the literature.

Response: Thank you for your question. We didn’t find literature demonstrating that LDRT can decrease tumor-associated antigens. But we agree with the reviewer’s idea since the sentence ‘HDRT is superior to LDRT in ‘inspire’ immune system via various immune pathways that trigger recognition and presentation of tumor-associated antigens.’ is not exact. We have changed it into ‘HDRT is likely to ‘inspire’ the immune system via various immune pathways that trigger recognition and presentation of tumor-associated antigens.’ in the last paragraph of page 4.

Comment 9: There is a formatting problem here; the figure legend is presented in the body text.

Response: We are very sorry for our incorrect layout and have corrected it on page 5.

Comment 10: But this comes upstream to the previous mechanisms; please change and make the order logical.

Response: We have corrected it in the figure 1 legend. ‘Stage2. HDRT Stimulates anti-tumor immunity (2). After high-dose irradiation, many immune pathways are activated. 1) HDRT will cause tumor cell necrosis and release of tumor cell debris, which can be taken up by antigen-presenting cells (APCs). 2) HDRT causes dsDNA release, triggering the cGAS-STING-IFN-1 pathway. IFN-1 recruits DCs that cross-prime CD8+ T cells. 3) HDRT upregulates cell surface molecules such as MHC-1/Fas/NKG2D, enhancing recognition and cytolysis of tumors by T cells and natural killer (NK) cells, respectively. Activated APCs then migrate to lymph nodes, where they educate and prime cytotoxic T-cells (3). Cytotoxic T-cells will enter the bloodstream (4) to reach primary and distant tumor sites and kill tumor cells.’

Comment 11: ‘LDRT can be applied to all tumor sites (both the local tumor which has received HDRT and other tumor lesions), if feasible.’ Very unclear.

Response: Many preclinical studies apply LDRT to different sites, some of them delivered LDRT to the secondary tumor[19], and some of them delivered LDRT to the same tumor site which had previously received HDRT[26]. We change the sentence into ‘LDRT can be applied to the local tumor which has received HDRT or other tumor lesions, which remains to be explored by future studies.’ in the figure 1 legend.

[19] H. B. Barsoumian et al., Low-dose radiation treatment enhances systemic antitumor immune responses by overcoming the inhibitory stroma. J Immunother Cancer 8, (2020).

[26] T. Savage, S. Pandey, C. Guha, Postablation Modulation after Single High-Dose Radiation Therapy Improves Tumor Control via Enhanced Immunomodulation. 26, 910-921 (2020).

Comment 12: But why aren’t these cells inactivated by the LDRT? The radiosensitivity of immune cells is high, so a depletion is expected. Signifies cruciaI timing for when to deliver LDRT.

Response: It is really a great suggestion as the Reviewer pointed out that we need to signify the timing for when to deliver LDRT so as to avoid immune cell depletion. Some researchers found delivering LDRT(1Gy/d for 2 days) 3 days after HDRT increased immune cell infiltration and did not cause immune cell depletion[19], some observed 0.5Gy*4 one day after HDRT increased infiltration of immune effector cells[26]. An agreement has not been reached about the crucial timing for when to deliver LDRT in order to avoid effector immune cells depletion, and future studies are warranted. We add ‘The crucial timing of LDRT needs to be explored by future studies in order to avoid effector immune cell depletion.’ in the figure 1 legend.

[19] H. B. Barsoumian et al., Low-dose radiation treatment enhances systemic antitumor immune responses by overcoming the inhibitory stroma. J Immunother Cancer 8, (2020).

[26] T. Savage, S. Pandey, C. Guha, Postablation Modulation after Single High-Dose Radiation Therapy Improves Tumor Control via Enhanced Immunomodulation. 26, 910-921 (2020).

Comment 13: Line 191 Stated almost identically in previous section.

Response: We have changed this sentence into ‘HDRT kills tumor cells directly and the tumor cell debris can be uptaken by APCs such as DCs.’ in the last paragraph of page 5.

Comment 14: A discussion on local vs whole body LDRT is highly warranted.

Response: Local LDRT has been shown to promote effector immune cells infiltration, polarize immune cells into anti-tumor phenotype, decrease immune-suppressive cells such as Tregs and reduce immune-suppressive factors such as TGF-β in the irradiated tumor site[19,26]. Whole body LDRT alteres the immunosuppressive tumor microenvironment and enhances systemic immune response induced by HDRT[16,67]. Both of them can reverse the immune-suppressive TME and have complementary effects with HDRT. However, whole body LDRT is prone to induce systemic immune response, while local LDRT’s immune effect is localized in the irradiated tumor site.

[19] H. B. Barsoumian et al., Low-dose radiation treatment enhances systemic antitumor immune responses by overcoming the inhibitory stroma. J Immunother Cancer 8, (2020).

[26] T. Savage, S. Pandey, C. Guha, Postablation Modulation after Single High-Dose Radiation Therapy Improves Tumor Control via Enhanced Immunomodulation. 26, 910-921 (2020).

[16]J. Liu et al., Low-Dose Total Body Irradiation Can Enhance Systemic Immune Related Response Induced by Hypo-Fractionated Radiation. Front Immunol 10, 317 (2019).

[67]R. Liu, S. Xiong, L. Zhang, Y. Chu, Enhancement of antitumor immunity by low-dose total body irradiationis associated with selectively decreasing the proportion and number of T regulatory cells. Cell Mol Immunol 7, 157-162 (2010).

Comment 15: ‘Felix Klug et al found LDRT(2Gy*1) reprogrammed TME by polarizing iNOS+M1 macrophages. In turn, iNOS-positive M1 macrophages produced relevant chemokines to recruit effector T cells, normalizes tumor vasculature, allowing T-cell infiltration.’ I cannot see any mechanism related to this statement ‘normalizes tumor vasculature’

Response: Irradiation triggered the polarization of M2-like toward M1-like iNOS expressing tumor-associated macrophages. iNOS activity by these reprogrammed macrophages was responsible for vascular normalization and activation, T cell recruitment, and tumor rejection. We have added ‘Klug et al. found that LDRT (2 Gy*1) reprogrammed the TME by polarizing M1 macrophages. In turn, iNOS activity by these reprogrammed macrophages was responsible for vascular normalization and activation, T cell recruitment, and tumor rejection’ in the second paragraph of page 7.

Comment 16: ‘Irradiation decreases vascular volume in a dose-dependent manner.’ This is not accurate; irradiation can, at least temporarily, increase vascular permeability, therby enhancing tissue perfusion.

Response: We made corrections to this sentence in the third paragraph of page 7: Irradiation decreases tumor vascular volume in a dose-dependent manner.

Comment 17: ‘ Four-month disease control rate (DCR, complete/partial response [CR/PR] or stable disease [SD]) was 42% (47% [16/34] in HDRT+LDRT group vs. 37% [14/38] in HDRT alone group, P = 0.38), and overall response (ORR, CR/PR at any point) was 19% (26% [9/34] in HDRT+LDRT group vs. 13% [5/38] in HDRT alone group, P = 0.27).’ Rewrite so that key figure appear more straightforward.

Response: We have rephrased the sentence: Primary endpoints were disease control rate(DCR) and overall response rate(ORR). DCR was defined as complete/partial response [CR/PR] or stable disease [SD], and ORR was defined as CR/PR at any point. Four-month DCR was 47% [16/34] in HDRT+LDRT group vs. 37% [14/38] in HDRT alone group, P = 0.38, and ORR was 6% [9/34] in HDRT+LDRT group vs. 13% [5/38] in HDRT alone group, P = 0.27.

Comment 18: Line 339: This is a little ‘back-to-basics’; SBRT of oligometastatic disease. In this case, tumor sterilization is ensured directly through irradiation. The authors must include this aspect.

Response: We added this point in the sixth paragraph of page 8: In addition, delivering SBRT to all tumor sites directly ensured tumor sterilization.

Comment 19: Line 351: Improved compared to what? Pivotal that comparison basis is provided.

Response: This is a single-arm phase 1 study so there is no control group[82]. We are very sorry for our misleading expression and have made corrections: ‘These data showed that multisite SBRT effectively controlled both the primary and distant tumor with acceptable toxicity in patients with metastatic cancer, demonstrating the safety and efficiency of multisite SBRT.’

[82]J. J. Luke et al., Safety and Clinical Activity of Pembrolizumab and Multisite Stereotactic Body Radiotherapy in Patients With Advanced Solid Tumors. J Clin Oncol 36, 1611-1618 (2018).

Comment 20: Line 357: ‘In conclusion, as long as feasible and tolerable, HDRT should be given to as many as possible tumor sites to achieve optimal tumor control and systemic immune response.’ This is self-evident due to the primary action of radiation, and has little to do with the immune system. The authors must greatly change improve their arguments here.

Response: We agree that all-site HDRT can have ablative effects on metastatic tumor and we have added ‘For one thing, delivering HDRT to all tumor sites ensure tumor sterilization; for another, multisite HDRT are prone to activate systemic immune response, which is crucial for eliminating circulating tumor cells that cannot be eliminated by HDRT.’ in the first paragraph of page 9.  

Comment 21: Line 382: However, this could also be a immunostimulatory reaction from irradiating the primary tumor nearby?

Response: Theoretically, irradiating the primary tumor nearby can induce an immunostimulatory reaction since the primary tumor will unintentionally receive scattered low-dose irradiation. However, this unintentionally scattered dose is uncontrollable in clinical practice, so we changed our wording according to reviewer’s suggestion in the forth paragraph of page 9: ‘scattered low-dose irradiation can be potentially utilized to enhance abscopal effect and tumor control’. Future studies are warranted before putting this scattered low-dose irradiation into clinical practice.

Comment 22: Line 428, Still, some studies show that partial misses of the tumor lead to local failure. Are these tumors immunologically cold? Please comment.

Response: The two study investigating abscopal effect of partial-volume HDRT used immunologically cold tumor models. We discussed this aspect in the second paragraph of page 10: Markovsky et al conducted an experiment in mice bearing bilateral breast tumors. They blocked one half of the volume from the primary radiation field to investigate whether partial-volume HDRT (10 Gy*1) is inferior to full-volume radiation in tumor control. Their data showed that partial HDRT led to an abscopal effect similar to fully exposed tumors [89]. When using the less immunogenic Lewis Lung Carcinoma mouse model to perform similar experiments, they found 15Gy was required to achieve adequate tumor response, confirming that the findings are not unique to 67NR breast cancer or otherwise highly immunogenic tumors. Similarly, Yasmin-Karim et al. treated mice bearing bilateral prostate tumors with one site whole-tumor HDRT (5 Gy*1) or partial-volume (the center zone of tumor) HDRT. They observed when using smaller field sizes, the abscopal effect, cytotoxic CD8+T-lymphocytes infiltration, and the positive shift of pro-inflammatory cytokines were equal to irradiating the whole tumor volume. Mice treated with partial-volume HDRT even experienced better survival [90].

[89] E. Markovsky et al., An Antitumor Immune Response Is Evoked by Partial-Volume Single-Dose Radiation in 2 Murine Models. Int J Radiat Oncol Biol Phys 103, 697-708 (2019).

[90] S. Yasmin-Karim et al., Boosting the abscopal effect using immunogenic biomaterials with varying radiotherapy field sizes. (2021).

Comment 23: Line 449: So what about fractionating SFRT? I.e. deliver it several times with lower max dose per fraction?

Response: Up to now, there is no study investigating the immune-stimulating effect of fractionating spatially fractionated radiation therapy(SFRT). However, from our point of view, fractionating SFRT might be inferior to delivering SFRT in a single high dose, because immune cell populations that are crucial for boosting the abscopal effect (such as APCs or CD8+cytotoxic T cells) are particularly radiosensitive. High radiation dose or repeated fractions of RT are likely to suppress or stifle their action needed to generate a robust abscopal effect. Future study are warranted to compare the immune-stimulating effect of different doses and fractions of SFRT.

Comment 24: ‘Some researchers divide HDRT into 2 kinds of dose according to the effects on immune system: immune priming dose and ablative dose. The former represents radiation dose that induces in situ vaccination (6-12Gy, because a dose >12-18Gy can induce TREX1, which clears the cytosolic DNA induced by HDRT and consequently impairing cGAS-STING-IFNγ pathway[94]), and the latter (a biologic effective dose (BED) in excess of 100 Gy) is designed to ablate all tumor and immune components.’ either state without a parenthesis or skip.

Response: We have made corrections according to reviewer’s request. ‘Some researchers divide HDRT into two kinds of doses according to their effects on the immune system: an immune-priming dose and an ablative dose. The former represents a radiation dose that induces in situ vaccination. Generally, it is defined as 6-12 Gy since a dose >12-18Gy can induce TREX1, which clears the cytosolic DNA induced by HDRT and consequently impairs the cGAS-STING pathway. The latter is defined as a biologic effective dose (BED) in excess of 100 Gy, which is designed to ablate all tumor and immune components.’

Comment 25: Line 466: Indeed, so how does patients’ immune function contribute to treatment decision making as noted in the previous paragraph?

Response: We included this aspect in Part 5: Clinical Decision Suggestions.

Comment 26: Figure 2, Does the bag signify immuntherapy?

Response: The bag siginifies adoptive cell therapy. We have made a mark in figure 2.

Reviewer 2 Report

Reviewer's report
Manuscript ID: Cancers-1774369
Title:
 A ‘Hybrid’ Radiotherapy Regimen Designed for Immunomodulation: Combining High-dose Radiotherapy with Low-dose  Radiotherapy 

Date:2022/6/17

Reviewer's report:
This is an interesting review article toward the benefit of combining high dose radiotherapy and low dose radiotherapy as an immune modulator in the treatment of cancer. Radiotherapy is used routinely as the standard of treatment for >50% of cancer.  Recent evidence indicates that ionizing radiation can enhance immune responses to tumors.  Hypofractionation or other modifications of standard fractionation mayimprove radiation’s ability to promote immune responses to tumors. However, there is limited understanding of the immunological impact of hypofractionated  (HDRT) and  (LDRT) regimens, as these observations are relatively recent. This MS was one the few study base on the currrent evidence supporting the use of HDRT + LDRT, and thoroughly explains possible immune-biology mechanisms of ‘hybrid’ radiotherapy  . I'm sure this study could help in the decision-making process and guide towards an optimal therapeutic strategy in the future treatment of cancers.

The MS is well prepared and containing a large amount of data.  Although theory of combining HDRT+LDRT has matured, there are still many question need further investigation prior clinical practice.  Nevertheless, it was still well written, thus, it must be published.

Typo error  → 1. line 325 trail → trial

                         2. line 518 activat_ ing → activating

Author Response

Dear Editor:

Thanks for your letter and for the reviewers’ comments concerning our manuscript entitled ‘A ‘Hybrid’ Radiotherapy Regimen Designed for Immunomodulation: Combining High-dose Radiotherapy with Low-dose Radiotherapy’(Manuscript ID:1774369). Those comments are all valuable and helpful for revising and improving our paper. We have studied all comments carefully and have made the conscientious correction. Revised portions are marked in red on the paper. The main correction in the paper and the responses to the reviewers’ comments are as follows.

Response to Reviewer 2:

Reviewer 2: Typo error  → 1. line 325 trail → trial

2.line 518 activat_ ing → activating

Response: Thank you very much for taking your time to review this manuscript. We have corrected our typo error according to your suggestions and we have finished our English editing using MDPI Editing (Editing ID: 46425). The revised manuscript is provided in the attached file. Thank you again for pointing out our mistake.

Round 2

Reviewer 1 Report

No further comments, the authors have addressed all my concerns appropriately. 

Author Response

Dear Reviewer:

Thank you again for reviewing our paper so carefully. We are grateful for your valuable feedback, which is quite helpful to improve the quality of our manuscript.